# Contents and Correlations of *N^ε^*-(carboxymethyl)lysine, *N^ε^*-(carboxyethyl)lysine, Acrylamide and Nutrients in Plant-Based Meat Analogs

**DOI:** 10.3390/foods12101967

**Published:** 2023-05-12

**Authors:** Shuang Fu, Yurong Ma, Yinan Wang, Chongzhen Sun, Feng Chen, Ka-Wing Cheng, Bin Liu

**Affiliations:** 1Shenzhen Key Laboratory of Food Nutrition and Health, Institute for Advanced Study, Shenzhen University, Shenzhen 518060, China; fushuang0322@163.com (S.F.); sfchenmme@163.com (F.C.); kwcheng@szu.edu.cn (K.-W.C.); 2Institute for Innovative Development of Food Industry, Shenzhen University, Shenzhen 518060, China; wyn_o3o@163.com; 3School of Public Health, Guangdong Pharmaceutical University, Guangzhou 510006, China; sunchongzhen@gdpu.edu.cn

**Keywords:** plant-based meat analog, CML, CEL, acrylamide, nutrient

## Abstract

High temperatures applied in the production of plant-based meat analogs (PBMA) lead to the occurrence of Maillard reactions, in which harmful compounds *N^ε^*-(carboxymethyl)lysine (CML), *N^ε^*-(carboxyethyl)lysine (CEL) and acrylamide are formed. However, little research has focused on these compounds in PBMA. In this study, the contents of CML, CEL and acrylamide in 15 commercial-sold PBMA were determined by an ultra-high performance liquid chromatograph coupled with a triple quadrupole tandem mass spectrometer (UHPLC-QqQ-MS/MS). Nutrients (protein, amino acids, fatty acids and sugars) which are related to the formation of these compounds were also studied. The results showed that CML, CEL and acrylamide contents were in the range of 16.46–47.61 mg/kg, 25.21–86.23 mg/kg and 31.81–186.70 μg/kg, respectively. Proteins account for 24.03–53.18% of PBMA. Except for Met + Cys, which is the limiting amino acid of most PBMA, all other indispensable amino acids met the requirements for adults. Besides, PBMA had more n-6 fatty acids than n-3 fatty acids. A correlation analysis showed that proteins and the profiles of amino acid and fatty acid had little influence on CML but significant influence on CEL and acrylamide. The results of the present study can be used as a reference to produce PBMA with higher amounts of nutrients and lower amounts of CML, CEL and acrylamide.

## 1. Introduction

The non-enzymatic reaction known as the Maillard reaction can occur under high temperature when amino groups on proteins, peptides or amino acids react with carbonyl groups on reducing sugars. With the occurrence of Maillard reactions, a group of compounds referred to as advanced glycation end-products (AGEs) are formed, including *N^ε^*-(carboxymethyl)lysine (CML), *N^ε^*-(carboxyethyl)lysine (CEL), pyrraline, methylglyoxal-derived hydroimidazolone isomers (MG-Hs), glyoxal-derived hydroimidazolone isomers (GO-Hs), methylglyoxal lysine dimer (MOLD), glyoxal lysine dimer (GOLD), pentosidine and argpyrimidine [1]. CML is considered a biomarker for AGEs, since its precursor, glyoxal, is abundant during the thermal processing of food, and it can be formed at any stage of glycoxidation or lipoxidation [2]. CEL is another widely studied AGE in food. The formation of AGEs is of great interest to researchers in food and medical sciences, since these compounds are not only formed during the heat treatment, processing and storage of food, but also in vivo under physiological conditions [1,3]. Human studies have shown that about 10% of the AGEs present in ingested foods are absorbed. Of that amount, only 30% is excreted into urine by humans with normal renal function, whereas this percentage decreases to less than 5% for those with severe renal disease. The remaining AGEs are retained in the human body [4]. The accumulation of AGEs in human tissues is involved in the development of atherosclerosis, neurodegenerative diseases, diabetes and renal injury, as well as other diseases [5]. Therefore, restricting the intake of diet-derived AGEs is considered to be an effective strategy to reduce the accumulation of AGEs, thereby inhibiting the pathogenesis of AGE-related disorders. Dietary AGE restriction in animals could prevent the development of atherosclerosis and renal injury, as well as reducing insulin resistance [5].

Acrylamide is another compound that can be formed during Maillard reactions; it has been considered a probable carcinogenic compound to humans. Studies conducted on rats and mice indicated that acrylamide has cancer risks, but the evidence was inconclusive in humans. For this reason, acrylamide is classified as a compound that is probably carcinogenic to humans (Group 2A) [6]. Its formation in food is occurs when a free amino acid, asparagine and reducing sugars are present at temperatures above 120 °C [7,8]. At present, there are no permissible limits for dietary acrylamide. Based on animal experiments, the European Commission proposed a benchmark dose level (lower confidence limit) (BMDL_10_) of acrylamide of 430 and 170 μg/kg body weight per day (bw/d) for inducing peripheral neuropathy and neoplastic effect, respectively [9]. Due to its potential carcinogenicity, the Food and Drug Administration (FDA) issued a guidance to growers, manufacturers and food service operators to reduce the acrylamide content in foods [10]. Similarly, the European Union also established mitigation measures and benchmark levels for reducing the contents of acrylamide in foods (EU2017/2158). 

In recent years, there has been increasing interest in plant-based meat analogs (PBMA). The production of PBMA not only satisfies the demand of vegetarians for high quality protein but also eases the environment problems associated with the excessive consumption of meat and the animal welfare problems associated with their slaughter. Extrusion is one of the most widely used technologies for the production of PBMA. During production, plant-based protein is extruded by a single or twin-screw extruder under low or high moisture conditions. Due to their high protein contents, soybean protein concentrate, soybean protein isolate and pea protein are commonly used for the production of PBMA [11]. Carbohydrates such as starch are usually added to enhance the texture and consistency of PBMA [12,13]. In addition, high temperatures are applied to denature proteins and form a meat-like texture. Under high temperature, the lysine and arginine of soybean protein can be grafted with reducing sugars degraded from carbohydrates, causing the glycosylation of soybean proteins [14]. AGEs and acrylamide have been identified in extruded pet food and soybean flour [15,16]. Similarly, AGEs in commercially sold and self-made PBMA were found by Deng et al. [17]. These results indicate that CML, CEL and acrylamide can be formed during the production of PBMA. However, studies on PBMA have mainly focused on the formation of fibrous structures, changes in protein and amino acids during processing, as well as the color, texture, taste, morphology and nutritional quality of the final products [11,12,13,18]. Few studies have paid attention to the detrimental compounds (CML, CEL and acrylamide) in PBMA [17,19,20,21], and those that have addressed this topic only determined the contents of CML, CEL or acrylamide in a few kinds of plant-based burgers. Therefore, in the present study, the contents of CML, CEL and acrylamide in 15 commercially sold PBMA were investigated. Furthermore, in order to explore the possible factors influencing their formation, the contents of protein, amino acids, fatty acids and sugars which are involved in the formation of AGEs and acrylamide were also determined. The results from this study can not only help us to better understand the concentrations of CML, CEL and acrylamide in PBMA but can also provide guidance for reducing their contents during the production of PBMA.

## 2. Materials and Methods

### 2.1. Materials

CML, CEL, CML-d_4_ and CEL-d_4_ with 98% purity were obtained from Toronto Research Chemicals Inc. (TRC, Toronto, ON, Canada). Acrylamide-2,3,3-d_3_ with 98% purity was obtained from Shanghai Macklin Biochemical Co., Ltd. (Shanghai, China). Acrylamide with 99% purity, HPLC grade formic acid and acetonitrile were obtained from Merck KGaA (Darmstadt, Germany), Thermo Fisher Scientific Inc. (Waltham, MA, USA) and Merck KGaA (Darmstadt, Germany), respectively. Amino acids standard solution was purchased from Waters Corporation (Waters, Milform, MA, USA). 

### 2.2. Sample Preparation 

In the present study, a total of 15 plant-based meat analogs were selected based on the types of products which are commonly sold on the market. PBMA in the form of meatballs, ground meat, steaks, burger patties, chicken breasts and sauced beef were purchased over the internet and coded with letters A to O. They were produced by Impossible Foods (Oakland, CA, USA), Beyond Meat (Columbia, MO, USA), V2 Food (Sydney, Australia), Hero Protein (Shanghai, China), Harvest Gourmet (Tianjin, China), PFI Foods Co., Ltd. (Shanghai, China), Protein Meat (Shanghai, China), OmniPork (Hongkong, China), Hong Chang Bio-Tech (Suzhou) Co., Ltd. (Suzhou, China), Whole Perfect Food (Shenzen, China) and Caixinxiang Co., Ltd. (Fuzhou, China). Samples were freeze-dried with a vacuum freeze drier (Scientz-10N, Ningbo Scientz Biotechnology Co., Ltd., Ningbo, China) and then ground to powder using a grinder (SD-JR05, Sande Electric Appliance Co., Ltd., Foshan, China). After sieving with a 40-mesh sieve, the fine powders were stored at –80 °C until further analysis.

### 2.3. Extraction and Determination of CML and CEL

Our analysis of protein-bound CML and CEL was based on previously reported methods with slight modifications [22,23]. Briefly, 0.2 g of accurately weighed sample was placed into a 50 mL tube. After the addition of 5 mL of n-hexane, the tube was thoroughly oscillated for 3 min and then centrifuged at 5000 r/min for 6 min. Subsequently, the supernatant was discarded. This defatting procedure was performed 3 times. After the sediment was evaporated to dryness under nitrogen at 45 °C, 7.5 mL of sodium borate buffer (0.2 mol/L, pH 9.2) and 3.75 mL of sodium borohydride (1 mol/L in 0.1 mol/L NaOH) were added to reduce the sample at 4 °C overnight (for about 12 h). Then, 4 mL of chloroform-methanol (2:1, *v*:*v*) was added to the precipitated proteins. After centrifuging at 5000 r/min for 10 min and discarding the supernatant, the precipitation operation was repeated. The precipitated protein was then hydrolyzed with 5 mL of HCl (6 mol/L) at 110 °C for 24 h. The protein hydrolysate was filtered with qualitative filter paper and diluted to 25 mL, of which 5 mL was freeze-dried. The freeze-dried powder was reconstituted with 2.5 mL of distilled water. Then, 62.5 μL of CML-d_4_ (1 mg/L) and CEL-d_4_ (1 mg/L) was added, making the final concentration of internal standards 25 μg/L. One milliliter sample solution was passed through an MCX cartridge (60 mg/3 cc, Waters, Milford, MA, USA) which was pre-activated with 3 mL of methanol and 3 mL of distilled water. Then, the cartridge was washed with 3 mL distilled water and 3 mL methanol. Finally, 5 mL of methanol (containing 5% aqueous ammonia, *v*/*v*) was used as an elution solution to elute the target compounds. The solution with target compounds was dried under nitrogen at 50 °C, reconstituted with 10 mL of distilled water, and filtered through a 0.22 μm filter. The filtered solution was used for analysis.

The analysis of CML and CEL was performed on a Waters ACQUITY UPLC H-Class chromatographic system (Waters, Milford, MA, USA) coupled with a XEVO TQ-XS tandem triple quadrupole mass spectrometer (UHPLC-QqQ-MS/MS) (Waters, Milford, MA, USA). A Waters ACQUITY HSS T3 column (1.8 μm, 2.1 mm × 100 mm) was used for the separation. The column temperature was set at 30 °C and the injection volume was 1 μL. Ultrapure water with 0.3% formic acid was used as mobile phase; the flow rate was 0.08 mL/min. The detection of parent and fragment ions of CML and CEL was performed using positive electrospray ionization (ESI+) in multiple reaction monitoring (MRM) mode. The ESI conditions applied were the same as those in the study of Cheng et al. [24]. The retention time, precursor ion, product ion, collision energy and cone voltage of CML and CEL, as well as their isotope-labeled internal standards, are summarized in Appendix A. Additionally, chromatograms of CML, CEL and their respective internal standards are shown in Appendix A. Mixed standard solutions of CML and CEL were prepared at concentrations of 2 to 1000 μg/L, in which the range the calibration curves had excellent linearity. The limit of detection (LOD) and limit of quantitation (LOQ) were determined at signal-to-noise ratios of 3 and 10, respectively. For CML, the calibration equation, LOD and LOQ were y = 0.692994 × x + 0.494217 (R^2^ = 0.9990), 0.37 and 1.12 μg/L, respectively. For CEL, they were y = 0.75529 × x + 0.0.402503 (R^2^ = 0.9996), 0.26 and 0.77 μg/L. Internal standards CML-d_4_ and CEL-d_4_ were added into the standard solutions at final concentrations of 25 μg/L. The results were expressed as mg/kg.

### 2.4. Extraction and Determination of Acrylamide

Our analysis of acrylamide was based on previously reported methods with slight modifications [25]. Briefly, 1.0 g of accurately weighed sample was placed into a 50 mL tube. After the addition of 10 μL of internal standard acrylamide-d_3_ (10 mg/L), the mixture was thoroughly mixed. Then, 10 mL of ultrapure water and acetonitrile was successively added into the mixture and thoroughly oscillated for 3 min after each addition. After ultrasonic treatment for 2 min, 4.0 g anhydrous MgSO_4_ and 0.5 g of NaCl were added. The tube was immediately sealed and vigorously shaken. After cooling to room temperature (20 °C), the tube was centrifuged at 10,000 r/min for 6 min. An aliquot (6 mL) acetonitrile layer was transferred to a 15 mL tube, dried in nitrogen, reconstituted with 2 mL of ultrapure water and 3 mL of n-hexane and homogeneously mixed. The tube was centrifuged at 5000 r/min for 6 min to remove the n-hexane layer. The water layer was filtered through a 0.22 μm filter and used for UPLC-QqQ-MS/MS analysis.

The UPLC-QqQ-MS/MS used for the analysis of acrylamide was the same as that used for CML and CEL. The column temperature was set at 30 °C and the injection volume was 1 μL. Water (A) and acetonitrile (B) were used as mobile phases. The mobile phase was set as follows: 0 min, 100% A; 2 min, 100% A; 2.01 min, 80% A; 4 min, 80% A; 4.01 min, 100% A; and 6 min, 100% A. The flow rate was 0.3 mL/min. The detection of parent and fragment ions of acrylamide was performed with positive electrospray ionization (ESI+) in multiple reaction monitoring (MRM) mode. The ESI conditions applied were the same those used for CML and CEL, as mentioned in Section 2.3. The retention time, precursor ion, product ion, collision energy and cone voltage of acrylamide and its isotope-labeled internal standard are summarized in Appendix A. Chromatograms of CML, CEL and their respective internal standards in a PBMA are shown in Appendix A. Acrylamide standard solutions in concentrations of 2 to 500 μg/L were prepared. Good linearity was observed in this range. The calibration equation, LOD and LOQ of acrylamide were y = 1.22092 × x + 0.997117 (R^2^ = 0.9971), 0.51 and 1.52 μg/L, respectively. Internal standard acrylamide-d_3_ was added into the working solutions at the final concentration of 100 μg/L. The result was expressed as μg/kg.

### 2.5. Determination of Protein, Amino Acids, Sugars and Fatty Acids

The protein content of the samples was determined by a Dumas azotometer (D200, Hanon Advanced Technology Group Co., Hong Kong, China). The amino acid (Asp, Thr, Ser, Glu, Gly, Ala, Cys, Val, Met, Ile, Leu, Tyr, Phe, Lys, His and Arg) content was determined according to the Chinese National Food Safety Standards GB 5009.124-2016 [26]. An amino acid analyzer (L-8900, Hitachi, Tokyo, Japan) equipped with an ion exchange column was used for the separation. The result was expressed as g/100 g and the sum content of these 16 kinds of amino acid was marked as the total amino acids content. Sugars (fructose, glucose, sucrose, maltose and lactose) were determined according to the GB 5009.8-2016 [27]. An HPLC (Alliance e2695, Waters, Milform, MA, USA) equipped with an NH2 column (5 μm, 250 mm × 4.6 mm; Agilent, USA) was used for the separation. The result was expressed as g/100 g. Fatty acid contents were determined according to GB5009.168-2016 [28]. A gas chromatographic system (GC-2030, Shimadzu, Kyoto, Japan) equipped with a flame ionization detector (FID) and a CP-Sill 88 capillary column (100 m × 0.25 mm × 0.20 μm, Angilent, Santa Clara, CA, USA) was used for the determination of fatty acid methyl ester contents. The result was expressed as g/100 g.

### 2.6. Statistical Analysis

All determinations were conducted in triplicate. Data were expressed as means ± standard deviations (SD) and analyzed by one-way analysis of variance (ANOVA) using SPSS 20.0 (IBM SPSS statistic software, v. 20.0, SPSS Inc., Chicago, IL, USA). Differences between group means were assessed by Tukey HSD test. Correlation analysis was performed by Pearson’s two-tailed test. A *p*-value less than or equal to 0.05 was considered statistically significant.

## 3. Results and Discussion

### 3.1. CML and CEL Contents of Plant-Based Meat Analogs

As shown in Table 1, the CML and CEL contents in the tested PBMA were in the range of 16.46–47.61 and 25.21–86.23 mg/kg, respectively. This was consistent with the results of Deng et al., who reported the contents of CML and CEL in raw PBMA [17]. However, another study reported much lower levels of CML and CEL in fried vegetarian burgers, i.e., 4.2 and 6.0 mg/kg, respectively [20], which may be attributed to the difference in ingredients and processing methods. For comparison, the CML and CEL contents in fresh raw pork were 32.52 and 10.70 mg/kg protein, respectively, i.e., lower than those in the PBMA of the present study when presented as mg/kg protein [29]. Similar contents were also reported by Chen and Smith [30]. Research on the CML contents in 257 foods found that cereals showed the highest average CML level, at 25.5 mg/kg, which was lower than the average content of CML in PBMA (32.64 mg/kg) in the present study [21]. These results indicate that the AGE content in PBMA is higher than in most foods, including raw meat. 

The higher levels of CML and CEL in PBMA may be attributed to the materials used for extrusion and the extrusion procedure. Soybean protein isolate (SPI) and pea protein isolate (PPI), in which CML and CEL were detected, were the most commonly used proteins materials in the production of PBMA [11,17]. CML and CEL in SPI and PPI were attributed to the high temperatures applied during production [17]. Moreover, the extrusion procedure of PBMA, which involves the extrusion of protein and other ingredients (corn starch for example) under high temperature and moisture conditions, may also result in the formation of CML and CEL [15,16]. It is worth noting that the samples used in this study were not cooked. The CML and CEL contents would be much higher after cooking, since heat treatment can induce their formation [22]. Therefore, in order to reduce the dietary intake of CML and CEL in PBMA, it is necessary to reduce their levels in the raw materials and optimize the production procedure and cooking methods.

### 3.2. Acrylamide Content of Plant-Based Meat Analogs

The formation of acrylamide is induced by high temperature (over 120 °C) and low moisture in the presence of asparagine and reducing sugars [7,8]. In this study, the acrylamide content in commercially sold PBMA was found to range from 31.81 to 186.70 μg/kg, with the average content at 68.55 μg/kg (Table 1). This was similar to the content found in ‘Cake and pastry’ (66 μg/kg), and ‘Other products based on cereals’ (68 μg/kg), but much higher than that in ‘Soft bread’ (42 μg/kg), which was reported to be the second greatest contributor to the total acrylamide exposure of toddlers, other children and adolescents [9]. Meanwhile, these three foods groups were found to be the main contributors to the total acrylamide exposure of adults, elderly and very elderly [9]. Besides, it is worth noting that the PBMA used in the present study were raw, and the cooking process would significantly promote the occurrence of Maillard reactions, leading to a higher content of acrylamide. Usually, plant-based food such as fried potato products, cookies, crackers, breakfast cereals and coffee have higher levels acrylamide than animal-based food [9,10]. However, similar contents of acrylamide in pan-grilled beef and soy-based burgers were reported, i.e., 73.40 and 75.88 μg/kg, respectively [19]. The wide variability in acrylamide content among different studies could be attributed to differences in the raw materials, manufacturing process and technologies. The European Commission proposed a BMDL_10_ of consumed acrylamide of 430 and 170 μg/kg bw/d for inducing peripheral neuropathy and neoplastic effect, respectively, but a large amount of PBMA needs to be consumed to reach these levels. Nonetheless, it is necessary to be aware of the formation of acrylamide in PBMA [9]. Based on animal evidence of the margin of exposure (MOE), which was calculated based on the levels of dietary exposure to acrylamide across surveys and age groups, concern has arisen about the neoplastic effects of acrylamide on toddlers and children [9]. Mitigation measures for the formation of acrylamide need to be implemented during the production of PBMA.

Acrylamide in PBMA probably forms during the extrusion process, since it was found that the extrusion of soybean at 130 °C for 20–30 s resulted in an increase in acrylamide from 22.36 to 62.62 μg/kg [31]. A similar result was also reported in extruded soybean flour [15]. Moreover, the presence of soybean flour probably had a promoting effect on the formation of acrylamide, as seen in the addition of soybean okara in cookies, which increased in acrylamide content from 361.88 to 588.84 μg/kg [15]. This was probably a result of the high contents of asparagine and reducing sugar, which are the predominant reactants of acrylamide, in soybean okara. According to the ingredient information of PBMAs provided by their manufactures, only one ground beef patty and one meatball, with acrylamide contents of 43.07 and 64.00 μg/kg, were produced using pea protein instead of soybean protein. 

### 3.3. Protein Content of Plant-Based Meat Analogs

The protein content of PBMA was found to vary widely, ranging from 24.03% to 53.18%, depending on the variety and producer (Table 2). A comparison with beef showed that beef burgers had a higher protein content (26.79%) than soy-based burgers (19.27%) [19]. However, when calculated as dry weight, the protein content in this soy-based burger (with 50.25% moisture content) was 38.73%, which was in the range (31.88–39.05%) of the burger patties used in present study. A higher protein content was reported in burger patties made of 80% lean beef, 93% lean beef and pork, as well as in products purchased from Impossible burger and Beyond burger, with values of 54.00, 66.79%, 48.41%, 41.01% and 42.69% dry weight, respectively [32]. The average protein content of different PBMA groups (150 PBMA consumed hot products in 2019 and 236 in 2021) was 14.68–21.33% wet weight [11]. Considering the moisture content, this finding is in line with our present results. Compared with traditional meat products, only plant-based burgers, schnitzel and mince had slightly lower protein contents [11]. These results indicated that PBMA are good protein substitutes for meat products, since they are comparable to those traditional meat products in terms of their protein content. Moreover, according to the digestible indispensable amino acid score (DIAAS), which is recommended by FAO as the best way to determine protein quality, pork meat was classified as excellent quality protein, while soybean and pea protein were classified as high quality and no quality protein, respectively [33].

### 3.4. Amino Acid Profile of Plant-Based Meat Analogs

The total content of 16 kinds of amino acids in PBMA determined in the present study was in the range of 21.89–47.68 g/100 g (Table 2), which was higher than that in beef burger (21.17 g/100 g) [34]. The amino acids with the highest and lowest contents in PBMA were Glu and Met, respectively (Appendix A). This was in agreement with the results reported by de Marchi et al. [18]. A comparison between plant-based and meat-based burgers showed significant difference in 5 out of 18 detected amino acids [18]. Plant-based burgers had higher levels of Cys and Glu than animal-based burgers, while exhibiting lower levels of Ala, Gly and Met [18]. In the present study, the content of Met in PBMA was in the range of 0.01–0.48 g/100 g (Appendix A), with the median at 0.04 g/100 g, which was lower than the content reported in beef burger (0.68 g/100 g) [34]. This was because sulfur amino acids (Met and Cys) are the limiting amino acids of soybean. The total amino acid content shown in Table 2 was lower than the protein content. One of the explanations for this difference was that the content of Trp, which was very low, as determined in a preliminary experiment, was not included in this study. Besides, the protein content which was determined by a Dumas azotometer was calculated by the total content of nitrogen. Compounds other than amino acids containing nitrogen may also be detected, indicating a higher content of protein compared to total amino acids. 

The amino acid score (AAS) is used to determine the effectiveness of absorbed dietary nitrogen in meeting the indispensable amino acid requirement at a safe level of protein intake [35]. According to the report of the WHO/FAO/UHU, the AAS is calculated by dividing mg of amino acid in 1 g test protein with mg of amino acid in a requirement pattern [35]. In the present study, the AAS of indispensable amino acid (expect for Try) was calculated. As shown in Figure 1, the AAS of indispensable amino acids (except for Met + Cys) in PBMA were all higher than 1.0, indicating that they can meet the indispensable amino acid requirement for adults at a safe level of protein intake. However, the sulfur amino acids Met and Cys were found to be the limiting amino acids in some PBMA, which was verified by previous reports [32,33]. In some samples, the AAS of Met + Cys was over 1; this was probably a result of the complementary effect of other ingredients. The first limiting amino acids in soybean protein indicated soybean protein to be a ‘good’ protein source instead of an ‘excellent’ source [32,33]. Due to the presence of limiting amino acids, it is still necessary to ingest multiple proteins or increase the intake of plant-based protein to meet the amino acid requirements, even though the protein contents in PBMA were comparable to those of traditional meat products.

### 3.5. Fatty Acid Profile of Plant-Based Meat Analogs

During the production of PBMA, plant oils are usually added, since they can not only provide meat-like sensory properties but can also acted as emulsifiers, plasticizers and lubricants [36]. According to the ingredient information on the packaging, only one plant-based ground pork sample was devoid of any added oil, while the remaining 14 samples contained an assortment of plant oils such as canola oil, sunflower oil, coconut oil or other kinds of plant oils. For the sample without oil added, extremely low fatty acid contents were found. Its total fatty acid (TFA), saturated fatty acids (SFA), monounsaturated fatty acids (MUFA), polyunsaturated fatty acids (PUFA), n-3 and n-6 contents were 1.91, 0.26, 1.15, 0.5, 0.05 and 0.44 g/100 g, respectively (Table 3). Meanwhile, in other samples, the TFA, SFA, MUFA, PUFA, n-3 and n-6 contents were in the range of 21.59–42.36, 3.57–21.93, 2.83–16.03, 1.61–26.01, 0.04–2.05 and 1.57–25.82 g/100 g, respectively. A comparison between samples with and without oil showed that the fatty acid profile of PBMA was influenced by the added plant oils. It is known that n-3 and n-6 fatty acids are essential fatty acids. Their contents as well as proportions play important roles in regulating inflammation and anti-inflammation, thereby contributing to homeostasis [18]. In the present study, the n-6/n-3 ratio was in the range of 2.65–220.83, indicating a higher n-6 fatty acid concentration in PBMA. This finding was consistent with that of a previous report [18]. The predominant fatty acids among n-3 and n-6 fatty acids were found to be α–linolenic acid and α–linoleic acid, respectively. Since the fatty acids in PBMA mainly originated from added plant oils, the higher n-6 was attributed to the higher contents of α–linoleic acid than α–linolenic acid in these oils.

The fatty acid profile of PBMA showed that oleic acid, linoleic acid and lauric acid were the predominant fatty acids in eight PBMA samples, accounting for 64.09–78.87% of total fatty acids, while oleic acid, linoleic acid and palmitic acid were the predominant fatty acids in the other five PBMA samples, accounting for 88.87–94.66% of the total fatty acids. This was because oleic acid, linoleic acid, palmitic acid and/or lauric acid were the main components of fatty acids in the commonly used plant-based oils (e.g., canola oil, sunflower oil and coconut oil) during PBMA production [37]. Due to the high contents of unsaturated fatty acids (the sum of MUFA and PUFA), the unsaturated fatty acid content in 13 samples was higher than that of SFA.

### 3.6. Sugar Content of Plant-Based Meat Analogs

In meat products, sugar is usually used by direct addition or as a main ingredient of cooking sauces to improve flavor. According to the ingredient information on the packaging, among the tested 15 PBMA, 9 samples contained added white granulated sugar or glucose, 3 contained added barley maltose powder and/or fruit/vegetable extracts and 2 samples did not contain any added sugars. In the present study, the contents of lactose, fructose, glucose, sucrose and maltose in PBMA were determined to investigate their relationships with CML, CEL and acrylamide. As shown in Table 4, lactose was not detected in any of the tested samples, and the total sugar content was in the range of 0.33–6.52 g/100 g. In a study conducted in 2019, the sugar contents in PBMA were reported to be in the range of 0.77–2.07 g/100, while in 2021, the range was 0.88–2.28 g/100 g [11]. It should be noted that samples in the present study were lyophilized, which may have resulted in higher sugar contents than those reported in other studies.

### 3.7. Correlation Analysis

As undesired and harmful compounds, CML, CEL and acrylamide can be formed during Maillard reactions in the presence of protein/amino acids and reducing sugars [1]. Foods with higher protein and amino acid contents normally contain more amine groups, which increases the occurrence possibility of Maillard reactions. Besides, lipid oxidation is closely interrelated with Maillard reactions due to the same intermediates being generated and interacting during food processing [38]. For example, the presence of linoleic acid promotes the formation of CML [38]. Therefore, in the present study, correlations between protein, amino acids, sugars, fatty acids and CML, CEL, acrylamide were investigated. Pearson’s analysis showed that protein content was positively correlated with CEL content (Table 5). Previous research found strong correlations between CML, CEL and protein content in canned fish [39]. However, controversial results were found in sterilized meat [23]. The difference between these findings was probably due to the difference in food matrix, cooking methods and production procedures. As for amino acids, the Ser, Glu, Cys and Phe contents were positively correlated with CEL (Table 5). Though both CML and CEL are formed from the glycation of lysine on the amine group, no relationship was found between them and lysine.

Reducing sugars are another reactant in the formation of CML and CEL. In the present study, though sugars were added into most of tested samples, there was no significant correlation between sugars and CML/CEL, except for fructose, which was positively correlated with CML content (Table 6). Poor correlations between reducing sugars and AGEs were also found in the study of Zhao et al. (2021) [39]. The weak correlation between reducing sugars and AGEs was probably because the sugar added to most samples was white granulated sugar. The dominant ingredient of white granulated sugar is sucrose, which has been reported to have little effect on the formation of CML and CEL [40]. However, the addition of glucose, fructose and lactose increased the content of CML and CEL in heat-treated pork [40]. Therefore, ingredients containing high levels of reducing sugars should be avoided for the production of PBMA.

Lipid oxidation is another important pathway for the formation of CML and CEL, since dicarbonyl compounds generated during lipid oxidation such as glyoxal (GO) and methylglyoxal (MGO) could react with lysine, leading to the formation of CML and CEL [3,41]. A correlation analysis showed that neither individual fatty acids nor total fatty acids were significantly correlated with CML and CEL, while SFA was negatively correlated with CEL (Table 7), indicating that fatty acid profile has little influence on the formation of CML in PBMA. Similarly, during the sterilization of beef and pork, fat content was reported to have little effect on the formation of protein-bound CML and CEL [23]. However, the formation of CML and CEL could be accelerated by lipid oxidation in model system as well as in stored Chinese-style sausages [3]. This difference in research results was probably because the oxidation of lipids in our and Sun’s study was negligible [23], since the duration and temperature of heat treatments were short and low. Hence, the formation of CML and CEL was not significantly influenced.

Besides AGEs, acrylamide is another hazardous compound formed during Maillard reactions. A Pearson’s correlation analysis showed that the protein, Asp, Thr, Val, Arg and Lys contents in PBMA were all negatively correlated with the acrylamide content, indicating that an increase in the contents of protein and these amino acids could potentially inhibit acrylamide formation. The inhibitory effect of Lys and Trp on the formation of acrylamide was also reported in model systems [4,42]. Asn and Lys are the dominant amino acids involved in the formation of acrylamide and AGEs, respectively. Competition between Asn and Lys for dicarbonyl compounds may influence the formation of acrylamide as well as CML and CEL [42]. No significant correlation was found between sugars and acrylamide. However, a positive correlation was found between linoleic acid, PUFA, n-6 fatty acids, total unsaturated fatty acid (sum of MUFA and PUFA) and acrylamide, indicating that unsaturated fatty acids could induce the formation of acrylamide. Furthermore, the effect of unsaturated free fatty acids (UFFA) on acrylamide formation varied according to the pH of the reaction system [43]. The addition of UFFA induced the formation of acrylamide in systems with pH 4.6 but inhibited it at pH 6.0 [43]. Therefore, in order to minimize acrylamide formation in PBMA, ingredients such as ascorbic acid, which can provide acidic conditions, should not be considered during the production of PBMA. Meanwhile, reducing the additional amount of oil is also important to maintain a lower acrylamide content.

## 4. Conclusions

In conclusion, this study found that raw PBMA contained a notable amount of CML, CEL and acrylamide, with concentrations ranging from 16.46–47.61 mg/kg, 25.21–86.23 mg/kg and 31.81–186.70 μg/kg, respectively. The protein content of PBMA was in the range of 24.03–53.18%. Except for Met + Cys, the AAS of other indispensable amino acids were all over 1, indicating that Met + Cys was the limiting amino acid of most PBMA. To meet the amino acid requirements of humans, multiple plant proteins with high Met + Cys contents can be used for the production of PBMA. Intake of multiple proteins or increasing the intake of PBMA is recommended. Additionally, the higher n-6 than n-3 fatty acid content makes PBMA a healthier food for humans. Our Pearson’s correlation analysis showed that CML was positively correlated with fructose and CEL was positively correlated with protein, Ser, Glu, Cys and Phe but negatively correlated with saturated fatty acids. Acrylamide was negatively correlated with protein, Asp, Thr, Val, Arg and Lys but positively correlated with linoleic acid, PUFA, n-6 fatty acids and unsaturated fatty acids. In general, protein, the profile of amino acid and fatty acid were shown to have little influence on CML but significant influence on CEL and acrylamide. Future studies will focus on the formation mechanisms and reducing strategies of CML, CEL and acrylamide in PBMA.

## Figures and Tables

**Figure 1 foods-12-01967-f001:**
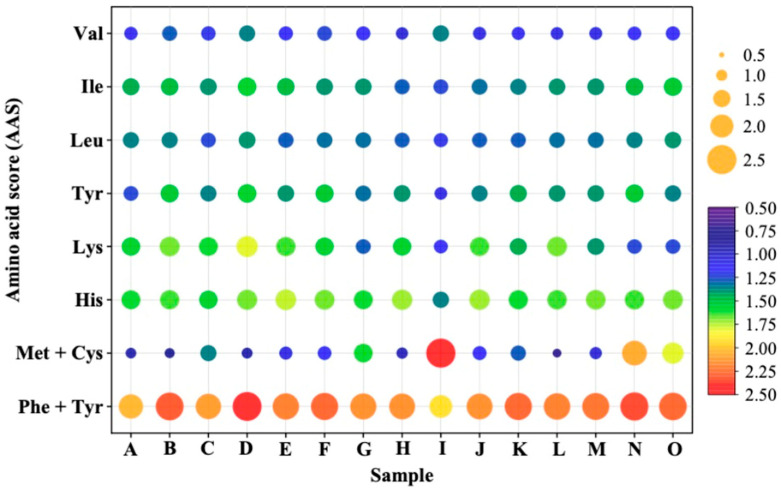
Amino acid score (AAS) of indispensable amino acid in plant-based meat analogs (A–O), as calculated according to the WHO/FAO/UNU (2013) recommended requirement patterns for adults.

**Table 1 foods-12-01967-t001:** CML, CEL and acrylamide contents of plant-based meat analogs.

Sample Code	CML (mg/kg)	CEL (mg/kg)	Acrylamide (μg/kg)
A	22.21 ± 1.12 ^ef^	39.97 ± 1.71 ^efg^	31.81 ± 0.26 ^f^
B	43.03 ± 2.87 ^ab^	35.45 ± 1.93 ^fgh^	43.07 ± 0.53 ^e^
C	22.47 ± 2.18 ^ef^	31.39 ± 2.20 ^hi^	33.59 ± 2.38 ^f^
D	40.59 ± 1.01 ^bc^	29.47 ± 0.41 ^hi^	64.00 ± 0.33 ^d^
E	34.05 ± 2.03 ^d^	51.97 ± 1.92 ^c^	63.07 ± 6.07 ^d^
F	47.61 ± 3.82 ^a^	46.25 ± 4.78 ^cde^	63.00 ± 1.47 ^d^
G	46.05 ± 1.46 ^ab^	86.23 ± 2.11 ^a^	69.00 ± 0.62 ^d^
H	22.79 ± 0.46 ^e^	42.32 ± 0.80 ^def^	49.07 ± 2.25 ^e^
I	41.67 ± 0.87 ^ab^	29.02 ± 1.70 ^hi^	66.00 ± 0.29 ^d^
J	43.42 ± 2.54 ^ab^	65.61 ± 4.13 ^b^	43.41 ± 0.50 ^e^
K	22.69 ± 0.48 ^e^	48.63 ± 4.43 ^cd^	78.63 ± 5.58 ^c^
L	25.92 ± 1.78 ^e^	25.21 ± 3.12 ^i^	48.96 ± 1.40 ^e^
M	34.53 ± 0.79 ^cd^	30.40 ± 0.83 ^hi^	120.78 ± 5.40 ^b^
N	26.16 ± 3.97 ^e^	71.56 ± 4.96 ^b^	67.22 ± 2.99 ^d^
O	16.46 ± 0.67 ^f^	31.94 ± 1.50 ^ghi^	186.70 ± 1.26 ^a^

Values are expressed as mean ± standard deviation (SD) (*n* = 3). Values in the same column marked by different letters indicate statistically significant differences (0.05 level).

**Table 2 foods-12-01967-t002:** Total amino acid and protein contents of plant-based meat analogs (g/100 g).

Sample Code	Total Amino Acids	Protein
A	37.18 ± 0.98 ^e^	39.05 ± 0.32 ^gh^
B	39.87 ± 1.06 ^cd^	43.56 ± 0.36 ^de^
C	38.83 ± 0.54 ^cde^	40.76 ± 0.08 ^f^
D	38.7 ± 0.74 ^de^	40.33 ± 0.12 f^g^
E	40.55 ± 0.82 ^cd^	42.81 ± 0.69 ^e^
F	41.21 ± 1.10 ^c^	45.15 ± 0.45 ^d^
G	34.28 ± 0.94 f	37.62 ± 0.33 ^h^
H	34.35 ± 0.71 ^f^	37.81 ± 0.67 ^h^
I	21.89 ± 0.4 ^h^	24.03 ± 0.21 ^k^
J	47.68 ± 2.19 ^a^	53.18 ± 0.39 ^a^
K	46.33 ± 1.14 ^ab^	49.38 ± 0.73 ^b^
L	30.77 ± 0.64 ^g^	33.7 ± 0.29 ^i^
M	30.34 ± 0.68 ^g^	31.88 ± 0.34 ^j^
N	45.09 ± 0.49 ^b^	47.21 ± 1.30 ^c^
O	23.63 ± 0.21 ^h^	24.18 ± 0.83 ^k^

Values are expressed as mean ± standard deviation (SD) (*n* = 3). Values in the same row marked by different letters indicate statistically significant differences (0.05 level).

**Table 3 foods-12-01967-t003:** Fatty acid contents of plant-based meat analogs.

Sample Code	SFA (g/100 g)	MUFA (g/100 g)	PUFA (g/100 g)	TFA (g/100 g)	n-3 (g/100 g)	n-6 (g/100 g)	n-6/n-3
A	10.36 ± 0.08 ^f^	15.41 ± 0.04 ^bc^	6.91 ± 0.03 ^f^	32.68 ± 0.15 ^de^	1.89 ± 0.01 ^b^	5.00 ± 0.02 ^ef^	2.65 ± 0
B	13.17 ± 0.22 ^c^	15.87 ± 0.25 ^ab^	6.73 ± 0.16 f^g^	35.77 ± 0.61 ^c^	1.41 ± 0.03 ^ef^	5.29 ± 0.12 ^ef^	3.76 ± 0.01
C	20.47 ± 0.27 ^b^	11.75 ± 0.14 ^e^	1.61 ± 0.02 ^i^	33.83 ± 0.42 ^d^	0.04 ± 0 ^k^	1.57 ± 0.02 ^h^	41.12 ± 0.23
D	12.62 ± 0.1 ^d^	16.03 ± 0.15 ^a^	7.02 ± 0.07 ^f^	35.67 ± 0.32 ^c^	1.50 ± 0.02 ^d^	5.49 ± 0.06 ^e^	3.65 ± 0
E	6.94 ± 0.11 ^g^	13.96 ± 0.21 ^d^	6.05 ± 0.10 ^g^	26.95 ± 0.41 ^g^	1.34 ± 0.02 ^f^	4.70 ± 0.08 ^fg^	3.52 ± 0.01
F	11.17 ± 0.11 ^e^	13.66 ± 0.15 ^d^	6.54 ± 0.06 ^fg^	31.37 ± 0.31 ^e^	1.44 ± 0.01 ^de^	5.07 ± 0.05 ^ef^	3.51 ± 0.01
G	5.21 ± 0.13 ^i^	8.31 ± 0.18 ^g^	20.51 ± 0.43 ^c^	34.02 ± 0.74 ^d^	1.68 ± 0.05 ^c^	18.81 ± 0.38 ^c^	11.19 ± 0.08
H	10.11 ± 0.23 ^f^	14.95 ± 0.39 ^c^	6.96 ± 0.18 ^f^	32.02 ± 0.81 ^e^	1.50 ± 0.04 ^d^	5.44 ± 0.15 ^e^	3.64 ± 0
I	4.18 ± 0.07 ^k^	7.31 ± 0.13 ^h^	16.08 ± 0.28 ^e^	27.57 ± 0.47 ^g^	1.16 ± 0.02 ^g^	14.91 ± 0.25 ^d^	12.87 ± 0.04
J	0.26 ± 0 ^m^	1.15 ± 0.01 ^k^	0.5 ± 0.01 ^j^	1.91 ± 0.02 ^i^	0.05 ± 0 ^k^	0.44 ± 0.01 ^i^	8.23 ± 0.08
K	12.70 ± 0.02 ^d^	2.83 ± 0 ^j^	6.06 ± 0.01 ^g^	21.59 ± 0.03 ^h^	0.78 ± 0 ^h^	5.27 ± 0.01 ^ef^	6.79 ± 0.01
L	21.93 ± 0.16 ^a^	6.17 ± 0.04 ^i^	4.44 ± 0.04 ^h^	32.54 ± 0.25 ^de^	0.32 ± 0 ^i^	4.11 ± 0.04 ^g^	12.75 ± 0.06
M	6.36 ± 0.15 ^h^	9.59 ± 0.23 ^f^	23.93 ± 0.57 ^b^	39.88 ± 0.94 ^b^	2.05 ± 0.05 ^a^	21.86 ± 0.52 ^b^	10.64 ± 0.05
N	3.57 ± 0.05 ^l^	7.18 ± 0.12 ^h^	19.10 ± 0.26 ^d^	29.85 ± 0.43 ^f^	0.09 ± 0 ^k^	19.00 ± 0.25 ^c^	220.83 ± 0.77
O	4.78 ± 0.06 ^j^	11.56 ± 0.15 ^e^	26.01 ± 0.36 ^a^	42.36 ± 0.57 ^a^	0.18 ± 0 ^j^	25.82 ± 0.36 ^a^	141.57 ± 0.56

SFA: saturated fatty acids; MUFA: monounsaturated fatty acids; PUFA: polyunsaturated fatty acids; TFA: total fatty acids; n-3: n-3 fatty acids; n-6: n-6 fatty acids. Values are expressed as mean ± standard deviation (SD) (*n* = 3). Values in the same column marked by different letters indicate statistically significant differences (0.05 level).

**Table 4 foods-12-01967-t004:** Sugar contents of plant-based meat analogs (g/100 g).

Sample Code	Fructose	Glucose	Sucrose	Maltose	Total Sugars
A	ND	1.44 ± 0.04 ^a^	0.80 ± 0.04 ^e^	ND	2.24 ± 0.03 ^g^
B	0.33 ± 0.01 ^e^	ND	ND	ND	0.33 ± 0.01 ^k^
C	ND	ND	0.53 ± 0.01 ^f^	ND	0.53 ± 0.01 ^j^
D	0.45 ± 0.02 ^bc^	ND	ND	0.53 ± 0.01 ^d^	0.98 ± 0.02 ^i^
E	0.50 ± 0.03 ^a^	0.47 ± 0.01 ^f^	0.58 ± 0.01 ^f^	1.00 ± 0.01 ^c^	2.55 ± 0.04 ^f^
F	0.42 ± 0.01 ^cd^	0.51 ± 0.02 ^f^	ND	0.51 ± 0.01 ^d^	1.44 ± 0.03 ^h^
G	0.52 ± 0.02 ^a^	1.28 ± 0.01 ^b^	ND	1.47 ± 0.03 ^b^	3.28 ± 0.05 ^d^
H	ND	ND	0.52 ± 0.01 ^fg^	ND	0.52 ± 0.01 ^j^
I	0.50 ± 0 ^a^	1.15 ± 0.01 ^c^	7.66 ± 0.01 ^a^	ND	9.31 ± 0.02 ^a^
J	ND	0.47 ± 0.01 ^f^	2.63 ± 0.02 ^b^	ND	3.10 ± 0.01 ^e^
K	ND	0.48 ± 0.01 ^f^	0.46 ± 0.02 ^g^	ND	0.94 ± 0.03 ^i^
L	0.38 ± 0.02 ^de^	0.51 ± 0.02 ^f^	2.57 ± 0.01 ^b^	ND	3.46 ± 0.04 ^c^
M	0.48 ± 0.01 ^ab^	1.03 ± 0.03 ^d^	2.26 ± 0.01 ^c^	2.74 ± 0.15 ^a^	6.52 ± 0.19 ^b^
N	0.33 ± 0.01 ^e^	0.74 ± 0.01 ^e^	0.46 ± 0.01 ^g^	ND	1.53 ± 0.03 ^h^
O	ND	ND	1.57 ± 0.04 ^d^	ND	1.57 ± 0.04 ^h^

ND: not detected. Values are expressed as mean ± standard deviation (SD) (*n* = 3). Values in the same column marked by different letters indicate statistically significant differences (0.05 level).

**Table 5 foods-12-01967-t005:** Correlations between protein, amino acids and CML, CEL and acrylamide.

Amino Acids/Protein	Correlation Coefficient (*r*)		
	CML	CEL	Acrylamide
Asp	0.204	0.349	−0.608 *
Thr	0.216	0.468	−0.556 *
Ser	0.158	0.535 *	−0.492
Glu	−0.110	0.550 *	−0.391
Gly	0.066	0.432	−0.326
Ala	−0.118	0.393	−0.233
Cys	0.096	0.520 *	−0.286
Val	0.390	0.440	−0.542 *
Met	−0.006	0.509	0.285
Ile	0.201	0.468	−0.459
Leu	0.220	0.501	−0.489
Tyr	0.215	0.444	−0.426
Phe	0.203	0.517 *	−0.430
Lys	0.240	0.223	−0.614 *
His	0.201	0.513	−0.474
Arg	0.291	0.353	−0.550 *
TAA	0.143	0.497	−0.500
Protein	0.207	0.519 *	−0.537 *

CML: *N^ε^*-(carboxymethyl)lysine; CEL: *N^ε^*-(carboxyethyl)lysine; TAA: total amino acids. * Correlation is significant at *p =* 0.05 (two-tailed).

**Table 6 foods-12-01967-t006:** Correlations between sugars and CML, CEL and acrylamide.

Sugars	Correlation Coefficient ^®^		
	CML	CEL	Acrylamide
fructose	0.652 **	0.075	0.007
glucose	0.204	0.339	−0.091
sucrose	0.115	−0.312	0.102
maltose	0.332	0.123	0.309
total sugar	0.300	−0.137	0.163

CML: *N^ε^*-(carboxymethyl)lysine; CEL: *N^ε^*-(carboxyethyl)lysine. ** Correlation is significant at 0.01 level (two-tailed).

**Table 7 foods-12-01967-t007:** Correlations between fatty acids and CML, CEL and acrylamide.

Fatty Acids	Correlation Coefficient (*r*)		
	CML	CEL	Acrylamide
SFA	−0.285	−0.539 *	−0.363
TUFA	−0.070	−0.074	0.649 **
MUFA	−0.009	−0.357	−0.068
PUFA	−0.074	0.123	0.777 **
n-3	0.374	−0.065	−0.079
n-6	−0.108	0.130	0.792 **
TFA	−0.254	−0.423	0.411
n-6/n-3	−0.418	0.237	0.429

CML: *N^ε^*-(carboxymethyl)lysine; CEL: *N^ε^*-(carboxyethyl)lysine; SFA: saturated fatty acid; TUFA: total unsaturated fatty acid; MUFA: monounsaturated fatty acid; PUFA: polyunsaturated fatty acid; n-3: n-3 fatty acids; n-6: n-6 fatty acids; TFA: total fatty acid; n-6/n-3: the ratio of n-6 to n-3. * Correlation is significant at *p =* 0.05 (two-tailed). ** Correlation is significant at *p =* 0.01 level (two-tailed).

## Data Availability

Data is contained within the article or Appendix A.

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
