# Peer review of "Contents and Correlations of Nε-(carboxymethyl)lysine, Nε-(carboxyethyl)lysine, Acrylamide and Nutrients in Plant-Based Meat Analogs"

_foods, 2023, doi:10.3390/foods12101967_

Round 1

Reviewer 1 Report

Foods

 Manuscript ID: foods-2374894

 Title: Contents and correlations of Nε-(carboxymethyl)lysine, Nε-(carboxyethyl)lysine, acrylamide and nutrients in plant-based meat products

 Authors: Shuang Fu, Yurong Ma, Yinan Wang, Chongzhen Sun, Feng Chen, Ka-Wing Cheng, Bin Liu

 The paper entitled Contents and correlations of Nε-(carboxymethyl)lysine, Nε(carboxyethyl)lysine, acrylamide and nutrients in plant-based meat products" deals with the formation of certain Maillard reaction products (PBMPs) in the manufacture of plant-based products and the identification of factors that might influence their formation. The authors focused on the influence of the content of proteins, amino acids, fatty acids and sugars in the products on the formation of Nε-(carboxymethyl)lysine (CML), Nε-(carboxyethyl)lysine (CEL) and acrylamide. In this study, 15 commercially available products (meatballs, ground meat, steak, burger patty, chicken breast, and sauced beef) were examined.
In the section "Results and discussion", the authors presented the information on the levels of CML, CEL and acrylamide in the investigated products and pointed out possible reasons for the high levels of Maillard reaction products in the samples, such as the type of raw materials used (soybean or pea protein isolates), the production conditions (temperature, humidity), the presence of reducing sugars and asparagine. The authors then presented their results in relation to the content of proteins, amino acids, fatty acids and sugars in the products studied, as well as the correlation analysis related to the increase of CML, CEL and acrylamide in food.

It is well known that Millard products not only produce the compounds responsible for the attractive taste and smell, but can also cause many serious diseases such as diabetes, atherosclerosis, cancer or Alzheimer's disease, etc. For this reason, the paper "Contents and correlations of Nε-(carboxymethyl)lysine, Nε (carboxyethyl)lysine, acrylamide and nutrients in plant-based meat products" is valuable and provides interesting information, especially for food manufacturers who want to produce healthy, safe and nutritious food.

This work is suitable for the „Foods” journal's aims and scope.

The entire experiment was designed in a proper way. The manuscript has been carefully prepared according to the guidelines for authors.

I suggest that the following minor errors be corrected:

1. The term "in line with" is used six times in this paper. I recommend replacing it with another term. 2. Page 5, line 194, "Results and discussion", the word "discussion" should start with a capital letter.

3. Page 8, Table No. 3: The heading of the table should be corrected, i.e. the first line should indicate the fatty acids, the second line the unit.

4. Page 11, Tables No. 6 and 7 should be separated by the text.

5. Page 13, "References", reference number 13, the year 1994 should be written in bold.

 I accept the submitted manuscript after a minor revision (correction of the above-mentioned errors).

Reviewer 2 Report

The problem of harmful organic compounds, including some Millard reaction products, which occur in the diet, and how to protect their synthesis in food, belongs to current scientific issues which are vital for human health safety. Plant-based meat are likely to become the main source of protein in the near future, which is why it is important to assess their content of both nutrients and harmful to health. Below you will find the list of some minor remarks which I propose to take into consideration while preparing a revised version of the manuscript.   Abstract Please edit the abstract to also state the analytical methods used to determine AA, CML and CEL   Introduction Line 54 - I suggest adding the information that AA is classified as a compound probably carcinogenic to humans (group 2A).   Line82/ 83 - Please provide literature references for this “few research papers”.   Materials and Methods Line 93 - should be: “Acrylamide-2,3,3-d3 Line 94 - Please check if you definitely used “HPLC grade acrylamide”?   Paragraph 2.3 and 2.4 Authors refer the readers to other methodological publications [19,20] in line 114,  [21] in line 142 and [22] in line 150. However, it would be good to supplement the manuscript with some basic and important analytical details on the results of the methods validation used for the extraction and determination of CML, CEL and AA in plant-based meat samples. If possible, please also provide examples of chromatograms confirming the presence of the determined compounds in food samples.   Line 172 - Probably should be “µg/kg”, similar to the results given for AA in Table 1   Results and discussion Line 399 – 403 - Please cite the publication {Yu, 2018 # 773} in a manner appropriate to FOODs.  Probably should be [3]. Line 402 - Please also include the reference number after the cited author's name: Sun’s study [?].     Table S1 Please verify that the quantification ions are labeled correctly. In the publication cited by the Authors ([21], in line 142), the product ion m/z =130 was used to quantify the CML and CEL.

Reviewer 3 Report

The manuscript by Fu et al presents an investigation where 18 commercial plant-based meat analog products are analyzed for Maillard reaction products, acrylamide, fatty acids content (MUFA and PUFA), protein content, amino acid content and content of mono- and disaccharides. Numerous commercial plant-based meat analog products have recently emerged, and it is therefore relevant to obtain knowledge on the chemical properties and the occurrence of undesired chemical reactions in these new product types. However, as the study is based on commercial products without in information on exact ingredients and processing methods (including heat exposure), the study has a rather observational nature. Mechanistic knowledge is not really generated as the experimental design does not allow this. Some comments are provided below:

The term plant-based meat products is unclear, especially when the authors also refer to a specific meat type, eg. pork, as reader I become uncertain whether the product is 100% plant-based or not. I would suggest to use the term ‘meat analogs’ rather than ‘meat products’

Absolute concentrations of acrylamide are reported. However, the authors do not evaluate these in light of toxicological threshold values. This would be relevant and also to compare content with other food items identified to be main sources of acrylamide in our diet.

It remains unclear if amino acids content is total content or content of free amino acids only. But from the reported values it seems that the reported content of amino acids is total content. What are the causes of the discrepancy between total amino acids and protein content?

English language needs to be improved.

English language needs to be improved.

Round 2

Reviewer 3 Report

No further comments